# Genetic Characterization of Blood Group Antigens for Polynesian Heritage Norfolk Island Residents

**DOI:** 10.3390/genes14091740

**Published:** 2023-08-30

**Authors:** Stacie O’Brien, Rodney A. Lea, Sudhir Jadhao, Simon Lee, Shrey Sukhadia, Vignesh Arunachalam, Eileen Roulis, Robert L. Flower, Lyn Griffiths, Shivashankar H. Nagaraj

**Affiliations:** 1Centre for Genomics and Personalized Health, Queensland University of Technology, Brisbane, QLD 4059, Australia; s55.obrien@qut.edu.au (S.O.); rodney.lea@qut.edu.au (R.A.L.); sudhirshriram.jadhao@connect.qut.edu.au (S.J.); s318.lee@hdr.qut.edu.au (S.L.); shreysanjaybhai.sukhadia@hdr.qut.edu.au (S.S.); vignesh.arunachalam@hdr.qut.edu.au (V.A.); lyn.griffiths@qut.edu.au (L.G.); 2Clinical Services and Research, Australian Red Cross Lifeblood, Brisbane, QLD 4059, Australia; eroulis@redcrossblood.org.au (E.R.); rflower@redcrossblood.org.au (R.L.F.)

**Keywords:** Norfolk Island, blood group, blood antigens, transfusion medicine

## Abstract

Improvements in blood group genotyping methods have allowed large scale population-based blood group genetics studies, facilitating the discovery of rare blood group antigens. Norfolk Island, an external and isolated territory of Australia, is one example of an underrepresented segment of the broader Australian population. Our study utilized whole genome sequencing data to characterize 43 blood group systems in 108 Norfolk Island residents. Blood group genotypes and phenotypes across the 43 systems were predicted using RBCeq. Predicted frequencies were compared to data available from the 1000G project. Additional copy number variation analysis was performed, investigating deletions outside of RHCE, RHD, and MNS systems. Examination of the ABO blood group system predicted a higher distribution of group A1 (45.37%) compared to group O (35.19%) in residents of the Norfolk Island group, similar to the distribution within European populations (42.94% and 38.97%, respectively). Examination of the Kidd blood group system demonstrated an increased prevalence of variants encoding the weakened Kidd phenotype at a combined prevalence of 12.04%, which is higher than that of the European population (5.96%) but lower than other populations in 1000G. Copy number variation analysis showed deletions within the Chido/Rodgers and ABO blood group systems. This study is the first step towards understanding blood group genotype and antigen distribution on Norfolk Island.

## 1. Introduction

Modern transfusion practices rely on the accurate identification of patients’ blood group antigens, detection of clinically significant antibodies, and the ability to provide patients with the best-matched blood and blood products. Blood group antigens consist of proteins, glycoproteins, and carbohydrates on the surface of the red cell that dictates their immunogenicity and serological phenotype [1]. In response to antigenic stimulation, circulating lymphocytes and/or plasma cells produce antigen-specific antibodies [1,2]. Exposure to foreign antigens during pregnancy or after a blood transfusion can result in development of these antibodies through a process known as alloimmunization [3]. Alloimmunization occurs in an estimated 1–3% of the transfused population, but might be 30% higher in regularly transfused patients, e.g., chronic kidney disease patients [4,5,6].

Finding the best possible matching unit between donor and patient blood groups may reduce the incidence of alloimmunization and decrease the risk of adverse transfusion reactions. The International Society of Blood Transfusion (ISBT) currently recognizes 44 different red cell blood group systems, containing >345 red cell antigens, which are comprised of >1600 alleles [7]. These antigens are inherited and consist of polymorphic structures located on the surface of red blood cells. Genome sequencing analysis can characterize all blood group system antigens, overcoming limitations in traditional serology/molecular based testing [8].

While the importance of extensive blood group antigen characterization in larger populations has been previously highlighted, many isolated populations, including founder populations, remain overlooked [9]. Founder populations are populations with reduced genetic diversity and higher homozygosity, usually due to genetic bottlenecks associated with geographical and cultural isolation. As a result, variants that are rare in global populations may occur at higher frequencies in founder populations [10]. One such population includes Norfolk Island; an isolated, external territory of Australia, located approximately 1456 km east of Brisbane, with a current population of ~1800 residents [11]. Many islanders can trace their heritage back to the original founders—a small group comprised of HMS Bounty mutineers and their Tahitian wives [12]. Many rare blood group antigens are clinically significant. For example, the JK null allele within the Kidd (JK) blood group system, lacks both Jk(a) and Jk(b) antigens, and has been shown to play a role in both hemolytic transfusion reactions (HTRs) and hemolytic disease of the fetus and newborn (HDFN) [13]. We hypothesize that the strong Polynesian heritage link in Norfolk Island may show an increased prevalence of JK null alleles compared to Caucasian populations.

The physical isolation of Norfolk Island makes it difficult to maintain a strong health service for residents, as, unlike other remote healthcare services, external assistance is only available through commercial or charter flights [14]. Maintaining a strong health service in remote areas relies on the ability to recruit and maintain skilled healthcare workers, having the necessary infrastructure in place to provide health services and understanding the needs of the community [14]. Norfolk Island currently has a 24-bed hospital that provides 34 different outpatient services. In 2012, maternity services were scrapped due to an inability to recruit and retain staff with suitable obstetric skills. As a result, all pregnancies are now managed by the Australian public health system and births occur on the mainland. The number of births on the island is relatively low, with the 2016 census showing a total of 40 recorded births for Australian territories of Norfolk Island, Christmas Island, Keeling Island, and Jervis Bay territory [15]. A significant challenge in remote communities involves patient management during critical bleeding situations. On Norfolk Island, the Australian Red Cross Lifeblood operates a blood bank in the hospital, with 175 residents as registered blood donors [14]. When a patient is critically bleeding, a clinical decision must be made if the immediate correction of blood loss or anemia outweighs the potential risks associated with blood transfusions, such as HTRs and alloimmunization.

Previous studies have demonstrated that the Norfolk Island population may have an increased risk of cardiovascular disease and metabolic syndrome compared to the mainland Australian population [16,17]. These studies suggest that their Polynesian heritage likely contributes to the risk of cardiovascular disease in this population, with Polynesian populations having an increase in total triglycerides, body mass index, and higher blood pressure as compared to other populations [16]. Although no data pertaining to cancer incidence in Norfolk Island are available, evidence suggests that such incidence is increased in Polynesian populations [18]. Blood transfusions are often required for the management of cancer and kidney disease [19,20]. Indeed, contrary to popular belief, most blood products are used to help manage patients with chronic conditions rather than trauma patients, with the Australian Red Cross Lifeblood estimating that 34% of transfused blood in Australia are used for patients with cancer or blood diseases [21].

The characterization of blood group antigens within the Norfolk Island population may provide building blocks to understand unique health challenges with transfusion medicine for residents. Characterization of blood groups alongside assessment of blood requirements may help improve blood transfusion safety within the community, help identify and stratify the risk of HDFN and identify any potential rare blood donors. In this study we used whole genome sequencing (WGS) data from 108 Norfolk Island residents to investigate their red cell blood group genotypes and predict antigens and phenotypes.

## 2. Methods

### 2.1. Norfolk Island Population Sample Selection and Whole Genome Sequencing Data

The Queensland University of Technology Human Research Ethics Committee granted approval of this study (approval code 3923) and was approved 10/05/2021. WGS data were obtained from whole blood DNA samples obtained from 108 consenting subjects enrolled in the Norfolk Island Health Study as reported previously [16]. The samples were selected from a large multi-generational pedigree within the population. The subsample of 108 subjects was specifically selected based on each subject having a direct genetic link back to the original founders of the population. The purpose of this selection method was to gain a representative subsample of the Norfolk islander founder population. As such, there was a significant degree of relatedness among subjects. Participants in this study were 18 years and over and signed informed consent forms prior to study participation, with ethical clearance granted by the Griffith University Human Research Ethics Committee [17]. The sequencing was performed on Illumina HiSeq X sequencers at the Kinghorn Centre for Clinical Genomics. The sequencing protocol involved paired-end Illumina TruSeq Nano DNA HT libraries and v2.5 clustering and sequencing reagents.

### 2.2. Variant Calling and Annotation

Sequencing reads (FASTQ) were mapped to the GRCh38 reference genome using Burrows–Wheeler Aligner (BWA) to generate binary alignment mappings (BAM) for an average 20× coverage [22]. GATK best practices were followed for data pre-processing, variant calling, and variant filtering [23]. The BAMTrimmer tool within RBCeq [24] was used to calculate blood group gene coverage (Figure 1). The blood group encoding gene variants annotation/categorization was performed using ANNOVAR [25].

### 2.3. Prediction of Red Cell Genotypes and Phenotypes Using RBCeq

The RBCeq software [24] takes single nucleotide variations (SNVs) and insertions/deletions (indels) from 48 genes and calculates CNVs, to facilitate the profiling of 43 blood groups and two transcription factors. RBCeq initially scans each blood group for reference and alternative alleles in the input VCF file, the zygosity of detected reference–alternate alleles guides sub-allele selection. Furthermore, the selected reference/alternate/sub alleles are paired against each other. They are then assigned a score that aids in selecting the dominant allele pair. The reference and alternative alleles scanned by RBCeq are stored in a back-end database that is based on International Society of Blood Transfusion (ISBT) allele tables [7,24].

### 2.4. Copy Number Variation Analysis

Additional CNV analysis was performed, focused on double deletions in blood group genes. A GATK gCNV pipeline was followed with a set interval size of 1000 bp. The pipeline was multithreaded, with the genome split into 100 intervals, which were analyzed separately and recombined at the end of the analysis [23]. The individual CNV vcf files were merged into a multivcf file and a tab delimited file containing the copy numbers at every 1000 bp intervals were analyzed. To assess the pathogenicity of the CNVs, we used AnnotSV [26], which follows ACMG-ClinGen guidelines [27].

### 2.5. Comparison with 1000 Genomes (1000G) Dataset

We performed a comparative analysis for the predicted genotype and phenotype frequencies between our population dataset and the 1000G dataset. The 1000G project ran from 2008–2015 and is an open access database containing 2504 whole genome samples from 26 different population groups. These 26 populations have been grouped into 5 main population types (America, East Asia, South Asia, Africa, and Europe). These data provide an overview of common human genetic variation and consist of a combination of WGS and dense microarray phenotype. For the present study, WGS BAM and VCF files were accessed through the 1000G project FTP server. Blood group profiles for 108 Norfolk Island residents were compared with population data obtained from 1000G [28].

### 2.6. Statistical Analysis

We utilized two-sample proportion tests to compare the populations of the 1000G against the NI population. A statistical test was conducted only when the variant frequency was found to be equal to or greater than 10%. To account for multiple comparisons, we employed the Bonferroni test for *p*-value correction. A significance level of *p*-value < 0.0016 (0.05/30*—5 comparison for each variant) was deemed statistically significant.

## 3. Results

### 3.1. Prevalence of Predicted ABO Phenotype

In examining ABO genotypes and their predicted phenotypes, our results showed that the predicted distribution of A1 is slightly higher than the distribution of group O (45.37% and 35.19%, respectively) (Figure 2). This predicted A1 prevalence is comparable with the distribution observed in European populations (42.94%). Our analysis showed that five participants (4.63%) had a predicted A_weak_ phenotype, and one participant (0.93%) had a predicted A_weak_B phenotype. While the group O distribution in our NI population was similar to the distribution seen in European populations (35.19% and 38.97%, respectively), group B distribution was slightly lower (4.63% compared to 6.76%).

### 3.2. RHD Characterisation and Its Prevalence

The Rh system was analyzed next, comparing the frequencies between NI and European populations. The Rh(D)-negative phenotype is reported at a prevalence of ~15% in European populations [29], whereas we observed a predicted ~9% Rh(D)-negative prevalence in this NI population (Table 1). Two participants were found to be homozygous for the *RHD*01W.33* allele (weak D type 33). No genotypes with heterozygous variants for *RHD*01W.33* were observed.

### 3.3. Comprehensive Analysis of the Kidd Blood Group among Norfolk Islanders

We examined the distribution of Kidd variants and predicted phenotypes among Norfolk Island residents (Table 2). While we did not predict any JK null phenotypes, we did identify several variants within the JK system. Our analysis showed three participants heterozygous for the *JK*02N.17*/*JK*02* genotype, with a predicted phenotype of Jk(a−b+), and one participant with a heterozygous genotype of *JK*01N.19*/*JK*01* with a predicted phenotype of Jk(a+b−). We also identified three participants with the homozygous genotype of *JK*01W.01*/*JK*01W.01*, with a predicted phenotype of Jk(a+^w^b−) (Table 2).

### 3.4. Prediction and Distribution of High Frequency Antigens

Our analysis showed a number of SNVs associated with predicted high frequency antigens across several blood group systems, including Knops (KN), Scianna (SC), Pel (PEL), Colton (CO), H, Kell (KEL), Cartwright (YT), and Landsteiner–Wiener (LW). For the KN system, we detected 15 individuals with a homozygous genotype of *KN*01.-05*/*KN*01.-05*, leading to a predicted loss of the high frequency antigen Yk(a−). We also identified four individuals with a homozygous genotype of *KN*01.-08*/*KN*01.-08*, with a predicted loss of the high frequency antigen Sl3. For the H system, one participant was homozygous for the *FUT2*01W.02.01*/*FUT2*01W.02.01* genotype which results in the H+^w^ secretor phenotype.

Several heterozygous alleles were also observed within this population. In the SC system, three participants genotyped with *SC*01*/*SC*01.-07.* In the CO system, there were seven participants with a *CO*01.01*/*CO*02.01* genotype, predictive of Co(a+b+) phenotype. Within the KEL system, we identified a single participant with a *KEL*02*/*KEL*01.01* genotype and predicted K+k+ phenotype. Finally, within the YT system, we also identified seven participants with a *YT*02*/*YT*01* genotype and predicted Yt(a+b+) phenotype (Table 3).

### 3.5. Analysis of Copy Number Variations among Norfolk Island Residents

We performed a CNV analysis focused on homozygous deleted regions in blood group genes. Additional GATK-pipelines were performed to examine CNVs in blood group systems, except for RHD and MNS, revealing a deletion of exon 27 in the Chido/Rodger’s system at a frequency of 4.63% (Table 4). A similar deletion was observed in exon 28 at a frequency of 5.55%. In *C4B*, the complete deletion of exon 28 was detected at a frequency of 7.40%, which may translate to an impact in the Ch/Rg system and *C4B* expression. Other notable findings in our population included a 46 bp deletion within exon 7 of the ABO system at a frequency of 1.85%, which may affect the ABO phenotype in these individuals (Table 4).

### 3.6. Prevalence of Predicted Weak Phenotypes and Low Frequency Antigens among Norfolk Island Residents

We examined the incidence of variants encoding for weak and low frequency antigens in a number of clinically significant blood group systems (Table 5) and found one individual heterozygous for *ACKR1* c.265C>T, resulting in the Duffy genotype *FY*01*/*FY*02W.01* or *FY*02*/*FY*02W.01*. These data also revealed a heterozygous *LW*05*/*LW*07* variant encoding LW(a+b+) present in one individual from the Norfolk Island study population with a similar frequency to that observed in the 1000G, as well as a rare heterozygous variant in the KEL system *KEL*02*/*KEL*02.03* in one individual with a predicted phenotype of K−k+. For PEL, we identified a single individual with the homozygous genotype *ABCC4*01.02W*/*ABCC4*01.02W*, leading to a predicted weakening of the Pel antigen and a predicted phenotype of Ple^w^+.

## 4. Discussion

Previous studies have demonstrated that red cell antigen distributions vary between ethnic groups, suggesting that founder effects, gene flow, and natural selection contribute to the differences observed in descendent populations [31]. Alloimmunization complicates transfusion practices—contributing towards adverse outcomes in transfusions and HDFN complications. Our study is the first to carry out extensive characterization of the 43 blood group systems and to analyze associated trends within the isolated Norfolk Island community, with the goal to characterize the blood group antigens for this community.

The ABO system is the most clinically significant blood group system. Transfusion mismatches in this system are the most common cause of life-threatening immediate intravascular hemolysis [1]. In this study, we describe a similar predicted phenotypic distribution of group A1 among the Norfolk Island population (45.37%) to that observed in European populations (42.94%). It was hypothesized by Simmons that higher group A prevalence was reported in Micronesia, Melanesia, and Indonesia due to a lack of the B antigen [32]. These findings align with our data, given the traceable European and Polynesian heritage of Norfolk Islanders and decreased prevalence of variants encoding for B antigens within the population. Our study also predicted seven participants with A2 phenotype (6.48%), five participants with weak A (4.63%), and one participant with AweakB phenotype (0.93%). The identification of weak ABO subgroups is clinically significant, as they can result in discrepancies in serology-based testing, which in turn may contribute towards delays in obtaining blood products.

There are known challenges associated with providing equitable care for isolated and rural communities due to limitations on resources, services, and the recruitment and retention of a skilled workforce [33]. Whilst most antibodies in the Rh system can cause HDFN and HTRs, anti-D is considered the most clinically significant in this system as it has the highest immunogenicity [3]. Our data indicated a prevalence of *RHD*-negative caused by a complete deletion of the *RHD* gene (*RHD*01N.01*/*RHD*01N.01*) of 9.26%. We also detected two participants homozygous for *RHD*01W.33* (weak D type 33), which consists of a SNP at c.520G>A [7].

The clinically significant Kidd (JK) blood group system is present on chromosome 18 (18q11-q12), and is encoded by the *SLC14A1* gene [13]. The null phenotype, Jk(a−b−), is rare in all populations but is reported at a higher frequency in Polynesian populations (1.0–1.4%) [13,31]. The most common JK null variant that is detected in Polynesian populations is *JK*02N.01*, which is a splice site mutation of intron 5 of the *JK*B* allele [13,34]. While we did not identify any participants with this mutation, we identified three participants with the heterozygous *JK*02*/*JK*02N.17* genotype, leading to a predicted phenotype of Jk(a−b+). This variant has not been previously described at an increased frequency in Polynesian or European populations. However, a recent study has shown that the SNP associated with *JK*02N.17* (c.810G>A) and its impact on JK null may be overstated and should be revisited [35]. We did observe several JK variants that occurred at a higher rate compared to the European population (Table 2). We identified three participants with a homozygous genotype of *JK*01W.01*/*JK*01W.01*, encoding for Jk(a+^w^b−) phenotype. This allele is commonly seen in East Asian populations but is not commonly associated with Polynesian or European heritage [13]. We speculate that the JK null variants observed in the Norfolk Island population in the present study is attributable to the inheritance of JK null silencing mutations from Polynesian founders.

High incidence antigens are those that have a prevalence of >90%, but are usually present at >99% in populations. Patients lacking these antigens with alloantibodies will struggle to find compatible units due to the high prevalence within the donor population [35]. Our study showed a notable finding within the Knops system. Knops antibodies are high-titer, low avidity antibodies and are considered clinically insignificant as they rarely cause HTRs or HDFN, however, patients with these antibodies cause know interferences with serological assays [1,35]. The Knops blood groups system consists of three low-prevalence antigens (Knb, McCb, and Vil) and six high prevalence antigens (Sja, KCAM, Kna, MccA, Yka, and Sl3) [3]. Knops is encoded by the complement receptor type 1 gene (*CR1*) on chromosome 1q32.2. The reference allele *KN*01* encodes the common antigens Kn(a+), McC(a), Sl1+, Sl3+, KCAM, and DACY+ [36]. Our results showed 15 participants homozygous for *KN*01.-05*/*KN*01.-05*, which predicts a loss of the high incidence antigen Yka. The Yka antigen is found in 92% of the Australian population [37], making this a notable finding. Additionally, our study demonstrated one participant homozygous for the *ABCC4*01.02W*/*ABCC4*01.02W* genotype within the PEL system, predictive of a weakened PEL phenotype, with PEL expression reduced by 30% [38].

Whilst SNVs define most of the blood group antigens, some blood group antigen expressions are impacted by larger insertions/deletions [2,39,40]. Our CNV analysis extended beyond the RHD/RHCE and MNS systems and revealed an increased frequency of CNVs associated with the Chido/Rodger’s blood group system. This blood group system consists of 41 exons and 9 antigens located in the C4d regions of C4 complement protein [41]. The *C4A* and *C4B* genes are highly homologous, sharing 99.91% nucleotide sequences, with many CNVs being described in Caucasian populations [41,42]. Due to the high homology between *C4A* and *C4B* genes, a better approach might be needed for subsequent studies. Our CNV analysis detected deletions between exons 27 and 28 in both *C4A* and *C4B* genes among Norfolk Island residents. Several diseases have been associated with mutations within the C4a/b regions of complement, with systemic lupus erythematosus (SLE) being particularly strongly linked to such mutations. The prevalence of SLE within Polynesian communities is higher than in the Caucasian population, affecting 51–73 per 100,000 individuals as compared to 15–19 per 100,000, respectively [43]. The prevalence of SLE within the Norfolk Island community was not assessed as part of this study, however, given the increased incidence observed in Polynesia, its associations with CNVs in *C4A*/*C4B* may contribute to the high frequency of this type of variant within our dataset [43]. CNV analysis also uncovered deletions within ABO. The ABO system consists of seven exons and 77% of its coding function occurs on exons 6 and 7. Variation within these exons is the predominant determinant of the A or B antigen, and loss of coding capacity may affect the function of glycosyltransferase, which may affect antigen expression [44]. However, as serology was not performed, there is no way to confirm the effect of this ABO variation.

In summary, we examined the prevalence of blood group genotypes and predicted phenotypes in 43 blood group systems among 108 residents of Norfolk Island. These data showed an expected similar correlation between Norfolk Island and European blood group frequencies for the ABO blood group system. The heterozygous JK null alleles predicted among Norfolk Islanders suggest that the null phenotype has been diluted in this population, with traceability back to Polynesian ancestry still being possible. Whilst our dataset comprised a small number of individuals selected for their direct descent to the founding population of Norfolk Island, our study uncovered a number of uncommon and potentially clinically significant variants which require further investigation. Whilst RBCeq predicts the red cell phenotype based on genetic information, it is important to note that genotype does not always correlate to phenotype, and the lack of basic or extended serology to determine the concordance of phenotype predictions to actual blood group phenotypes is a recognized limitation of this study. Future studies with confirmatory serology may help amend blood group predictions for Norfolk Island residents and is recommended.

The accurate characterization of blood group antigens in conjunction with an assessment of blood requirements may help reduce transfusion reactions, assess risk of hemolytic disease of fetuses and newborns, and improve blood safety for this isolated population. Given that Norfolk Islanders exhibit an increased risk of metabolic syndrome, cardiovascular disease, and kidney disease, and are isolated from mainland Australia, such an analysis would be useful to facilitate improved donor–patient matching in transfusion medicine.

## Figures and Tables

**Figure 1 genes-14-01740-f001:**
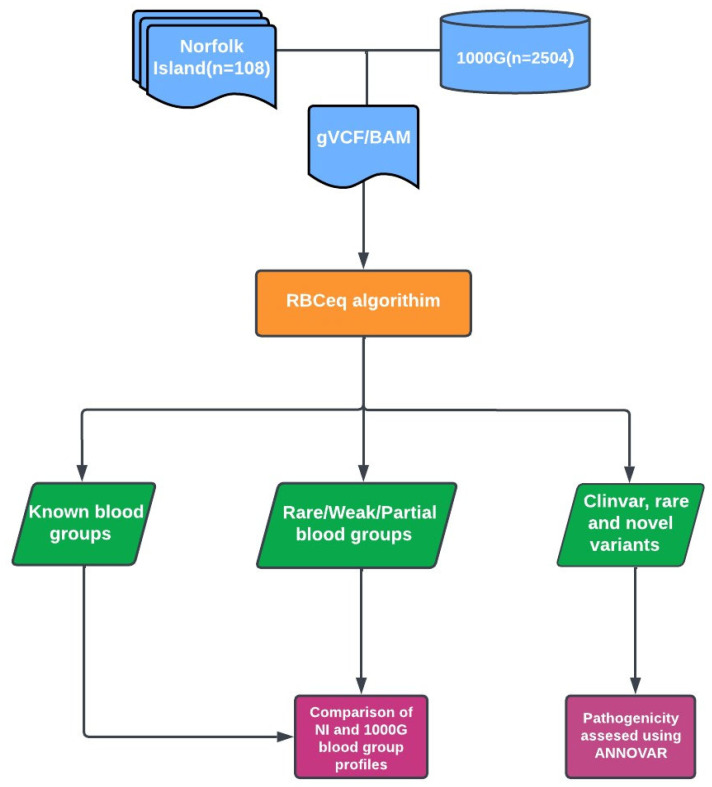
Overview of the workflow used to characterize population-specific blood group variants and predict phenotypes of Norfolk Island residents compared to populations in the 1000G. Blood group gene coverage was calculated using BAMstat and bedtools.

**Figure 2 genes-14-01740-f002:**
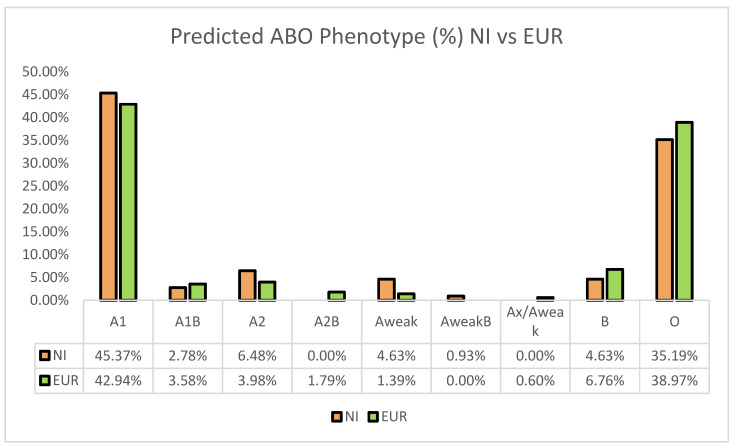
Comparison of ABO blood group frequencies in Norfolk Islander and European populations. Similar group A frequencies were observed when comparing Norfolk Islander (NI) and European populations, while group O prevalence was slightly lower among NI populations. Group B was seen at a reduced prevalence for Norfolk Island residents compared to European populations.

**Table 1 genes-14-01740-t001:** RHD predictions and variant calculations were compared to African, American, East Asian, European, and South Asian populations using data obtained from 1000G. We utilized two-sample proportion tests to compare the populations of the 1000G against the NI population showing statistically significant differences between NI and AFR and EAS.

Phenotype	Population	Frequency (%)	*p*-Value
D+	NI	96 (88.89)	-
	AFR	638 (96.52)	<0.0001 **
	AMR	326 (93.95)	0.119
	EAS	503 (99.80)	<0.0001 **
	EUR	422 (83.90)	0.245
	SAS	462 (94.48)	0.056
D-	NI	10 (9.26)	-
	AFR	23 (3.48)	0.013
	AMR	21 (6.05)	0.349
	EAS	1 (0.20)	NA
	EUR	81 (16.10)	0.096
	SAS	27 (5.52)	0.216
Weak D (type 33)	NI	2 (1.85)	-
	AFR	0 (0)	NA
	AMR	0 (0)	NA
	EAS	0 (0)	NA
	EUR	0 (0)	NA
	SAS	0 (0)	NA

**—Statistically significant results.

**Table 2 genes-14-01740-t002:** Analysis of the JK blood group system of Norfolk Island residents showed a slightly higher distribution of Jk(a+b+) variants compared to European populations. However, the number of weakened phenotypes Jk(a+^w^b−) was higher in NI compared to European, but lower compared to the other populations.

Phenotype	Population	Frequency (%)	*p*-Value
Jk(a+b−)	NI	37 (34.26)	-
	AFR	336 (50.83)	0.001
	AMR	72 (20.75)	0.006
	EAS	32 (6.35)	NA
	EUR	124 (24.65)	0.05
	SAS	155 (31.69)	0.688
Jk(a+b+)	NI	45 (41.67)	-
	AFR	126 (19.06)	<0.0001 **
	AMR	85 (24.49)	0.001
	EAS	46 (9.13)	<0.0001
	EUR	169 (33.59)	0.138
	SAS	99 (20.25)	<0.0001 **
Jk(a+b+^w^)/Jk(a+^w^b+)	NI	10 (9.26)	-
	AFR	0 (0)	NA
	AMR	0 (0)	NA
	EAS	2 (0.4)	NA
	EUR	0 (0)	NA
	SAS	0 (0)	NA
Jk(a−b+)	NI	13 (12.04)	-
	AFR	55 (8.32)	0.281
	AMR	102 (29.39)	<0.0001 **
	EAS	136 (26.98)	<0.0001 **
	EUR	129 (25.65)	0.004
	SAS	68 (13.91)	0.72
Jk(a+^w^b−)	NI	3 (2.78)	-
	AFR	62 (38.58)	NA
	AMR	22 (16.43)	NA
	EAS	80 (21.43)	NA
	EUR	19 (5.96)	NA
	SAS	57 (28.22)	NA

**—Statistically significant results.

**Table 3 genes-14-01740-t003:** Distributions of notable high incidence variants in the Norfolk Island population, predicting an increased prevalence of the Yka− antigen in the Knops blood group system, similar to the distribution seen in East Asian, American, and South Asian, and lower than European populations.

Blood Group System	ISBT & Nucleotide Change	Phenotype		RBCeq Prediction					
			Zygosity	NI% N = 108	AFR% N = 661	AMR% N = 347	EAS% N = 504	EUR% N = 503	SAS% N = 489
KN (ISBT 022)	*KN*01.-08* c4828T>A	Kn(a+b−) McC(a+b−) Sl1+ Yk(a+) Vil- Sl3- KCAM+	Heterozygous	5.56 (*n* = 6)	0.00 (*n* = 0)	1.15 (*n* = 4)	0.00 (*n* = 0)	3.77 (*n* = 20)	0.20 (*n* = 1)
KN (ISBT 022)	*KN*01-05* c.4223C>T	Kn(a+b−) McC(a+b−) Sl1+ Yk(a−) Vil- Sl3+ KCAM+	Homozygous	13.89 (*n* = 15)	0.30 (*n* = 2)	16.14 (*n* = 56)	12.70 (*n* = 64)	9.94 (*n* = 49)	10.84 (*n* = 53)
SC (ISBT 013)	*SC*01.-07*	Sc1+ SCAN+	Heterozygous	2.78 (*n* = 3)	0.00 (*n* = 0)	0.86 (*n* = 3)	0.00 (*n* = 0)	0.99 (*n* = 5)	0.20 (*n* = 1)
CO (ISBT 015)	*CO*01*/*CO*02* c.134C>T	Co(a+b+)	Heterozygous	6.48 (*n* = 7)	1.82 (*n* = 12)	1.73 (*n* = 6)	0.00 (*n* = 0)	6.56 (*n* = 33)	0.82 (*n* = 4)
KEL (ISBT 006)	*KEL*02* c.578C	K+k+	Heterozygous	3.70 (*n* = 4)	0.45 (*n* = 3)	4.32 (*n* = 15)	0.00 (*n* = 0)	7.55 (*n* = 38)	1.23 (*n* = 6)
YT (ISBT 011)	*YT*01*/*YT*02* c.1057C>A	Yt(a+b+)	Heterozygous	6.48 (*n* = 7)	0.00 (*n* = 0)	7.20 (*n* = 25)	0.40 (*n* = 2)	9.54 (*n* = 48)	8.38 (*n* = 41)

**Table 4 genes-14-01740-t004:** Copy number variation analysis for 108 Norfolk Island samples performed with GATK and gCNV and annotated by ANNOTSV [26]. This analysis was focused on double deletions in blood group genes and compared to global frequencies for reference.

Blood Group System/Gene	Genomic Positions for Deletions	Impacted Intron/Exon	Frequency NI	Frequency Global	OMIM ID	Clinical Significance
**CH/RG (ISBT 017)** * **C4A** *	31,995,858– 31,996,157	Intron 25-Intron 26	4.63%	6.51%	120810 [30]	C4a deficiency [30]
	31,996,158– 31,996,457	Intron 26-exon28	4.63%	6.51%	120810 [30]	C4a deficiency [30]
	31,996,458– 31,996,757	Exon 27-intron 28	5.55%	6.51%	120810 [30]	C4a deficiency [30]
**CH/RG** **(ISBT 017)** * **C4B** *	32,028,596– 32,028,895	Intron 25-Intron 26	5.55%	6.51%	120820 [30]	C4B deficiency [30]
	32,028,896– 32,029,195	Intron 26- Exon 28	7.40%	6.51%	120820 [30]	C4B deficiency [30]
	32,029,196– 32,029,495	Exon 28- Intron 28	7.40%	6.51%	120820 [30]	C4B deficiency [30]
**ABO** **(ISBT 001)**	133,255,603133,255,902	Exon 7- Exon 7	1.85%	1.29%	616093 [30]	Blood group, ABO system [30]

**Table 5 genes-14-01740-t005:** Notable variants of weak and rare phenotypical expressions in the Norfolk Island population and their frequencies compared to those observed in other populations including a prediction of the weakened PEL+w phenotype.

Blood Group System	ISBT	Zygosity	RBCeq Prediction						
			Phenotype	NI% N = 108	AFR% N = 661	AMR% N = 347	EAS% N = 504	EUR% N = 503	SAS% N = 489
FY (ISBT 008)	*FY*02W.01*	Heterozygous	Fy(a+b+^w^) OR Fy(a+^w^b+) OR Fyˣ	0.93 (*n* = 1)	0.00 (*n* = 0)	0.00 (*n* = 0)	0.00 (*n* = 0)	0.00 (*n* = 0)	0.00 (*n* = 0)
LW (ISBT 016)	*LW*05*/*LW*07*	Heterozygous	Lw(a+b+)	0.93 (*n* = 1)	0 (*n* = 0)	0 (*n* = 0)	0 (*n* = 0)	0.80 (*n* = 4)	0 (*n* = 0)
KEL (ISBT 006)	*KEL*02*/*KEL*02.03*	Heterozygous	K-k+	0.93 (*n* = 1)	0.30 (*n* = 2)	3.17 (*n* = 11)	0.00 (*n* = 0)	0.40 (*n* = 12)	0.20 (*n* = 1)
PEL (ISBT 040)	*ABCC4*01.02W* c.912G>T	Homozygous	PEL+^w^	0.93 (*n* = 1)	5.3 (*n* = 35)	0.29 (*n* = 1)	3.17 (*n* = 16)	0.99 (*n* = 5)	2.66 (*n* = 13)
H (ISBT 018)	*FUT2*01W.02.01*	Homozygous	H+^w^	0.93 (*n* = 1)	0.00 (*n* = 0)	0.00 (*n* = 0)	19.05 (*n* = 94)	0.00 (*n* = 0)	0.00 (*n* = 0)

## Data Availability

Due to ethical constraints, restricted data is in place. The Norfolk genetics steering committee will assess restricted data access requests via our GRC computational genetics group. Interested researchers should contact grccomputationalgenomics@gmail.com.

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
