# Peer review of "Genetic Characterization of Blood Group Antigens for Polynesian Heritage Norfolk Island Residents"

_genes, 2023, doi:10.3390/genes14091740_

Round 1

Reviewer 1 Report

The authors used whole genome sequencing data to describe the prevalence of 43 blood group systems among 108 Norfolk Island residents and applied the RBCeq algorithm to predict blood group phenotypes.

They concluded that their data showed an interesting correlation between Norfolk Island and European blood group frequencies for the ABO blood group system. The researchers also outlined how an accurate characterization of blood group antigens, along with assessment of blood requirements, can help reduce transfusion reactions, assess fetal and newborn hemolytic disease risk, and improve blood safety for this isolated population.

While the intent of the research is certainly important to help minimize risks and ensure healthier outcomes for both blood donors and recipients, especially in an isolated population such as Norfolk Islanders, the manuscript nevertheless needs major revision.

The major concerns are:

1)      the sample of 108 individuals is part of a large NI population study of 602 subjects; the authors refer to this work to describe their population, but a subsample is used in this work, and the authors do not describe how this subsample was chosen, why the sample size is so small, and what demographic characteristics this subsample has.

Sample description is extremely important for the reliability of allele frequency estimates.

2)      not even the slightest statistical analysis is done to test whether the differences observed by the authors between populations are significant or not. Even in a descriptive paper, a minimum of statistics must be there.

3)      the bibliography needs to be completely rewritten, because the references:

- 23 through 26, 32 through 34 are not cited in the text;

- number 14 is double

- sometimes the authors are missing or the journal is missing; and very often the order of authors; title; journal; vol/pages is random

- the numbering in the text goes up to number 18 and then goes to 35 in the materials and methods

- many references in the text are cited in the wrong place

The manuscript must be fully checked for spelling and typographical errors.

The manuscript denotes that it was very carelessly and sloppily written, and therefore does not reach the standards required for publication in the Genes journal in the present form.

Reviewer 2 Report

This is a valuable report on the distribution of blood group antigens in Norfolk Island with a current population of only 1,800 residents. It would be better with some additional information.

1. RBCeq showed that there were A1 (45.37%) and Jk weak (11.11%) in Norfolk Island, but did you confirm by serology that the RBCeq results were correct? If possible, please show the results of serology. If not,  please state the necessity of serological confirmation in discussion. 

2. Please briefly describe about the blood service in Norfolk in the introduction. It would be better to include basic information such as where and how blood products are transported or stocked, and how many products are supplied for patients each year.

3. Are the two participants with homozygous for RHD*01W.33 relatives?

4. Is the frequency of deletions between exons 27 and 28 of Chido/RG in Norfolk Island higher than in other ethnic groups? I would like to be listed the frequency of other ethnic groups in Table 4, if possible. The association between the prevalence of SLE and the frequency of CNVs in the CH/RG is interesting.

5. Reference #21 and #27 is same.
